# Comparative Study Effect of Different Urea Fertilizers and Tomato Pomace Composts on the Performance and Quality Traits of Processing Tomato (*Lycopersicon esculentum* Mill.)

**DOI:** 10.3390/plants13131852

**Published:** 2024-07-05

**Authors:** Ioanna Kakabouki, Ioannis Roussis, Magdalini Krokida, Antonios Mavroeidis, Panteleimon Stavropoulos, Stella Karydogianni, Dimitrios Beslemes, Evangelia Tigka

**Affiliations:** 1Laboratory of Agronomy, Department of Crop Science, Agricultural University of Athens, 11855 Athens, Greece; antoniosmauroeidis@gmail.com (A.M.); stavropoulosp@aua.gr (P.S.); karidogianni@aua.gr (S.K.); 2Laboratory of Process Analysis and Design, School of Chemical Engineering, National Technical University of Athens, 15780 Athens, Greece; mkrok@chemeng.ntua.gr; 3Institute of Industrial and Forage Crops, Hellenic Agricultural Organization Demeter, 41335 Larissa, Greece; dbeslemes@elgo.gr (D.B.); evitiga@yahoo.gr (E.T.)

**Keywords:** Heinz 3402 F_1_, nitrification and urease inhibitors, tomato pomace, organic fertilization, lycopene content

## Abstract

Processing tomato (*Lycopersicon esculentum* Mill.) is regarded amongst the most dominant horticultural crops globally. Yet, due to its elevated water and fertilization needs, its environmental footprint is significantly high. The recent efforts to reduce the footprint of agriculture have rekindled the search for optimized fertilization regimes in tomato. The aim of the present study was to assess the effect of different urea fertilizers and tomato pomace-based composts on the performance and quality traits of processing tomato. A two-year field experiment was conducted in the Larissa region, Central Greece, during 2018–2019. The experiment was set up in a randomized complete block design (RCBD), with five treatments: control, urea (Urea), urea with nitrification and urease inhibitors (Urea + NI + UI), processing tomato pomace with farmyard manure (TP + FM), and processing tomato pomace with compost from plant residues (TP + CM). Measurements included soil total nitrogen (STN), soil organic matter (SOM), root length density (RLD), arbuscular mycorrhiza fungi (AMF) colonization, dry weight per plant, fruit yield (number per plant, total yield, weight, diameter), fruit firmness, total soluble solids (TSS), titratable acidity (TA), lycopene content and yield, and fruit surface color (*L**, *a**, *b**, CI). Overall, the best results in soil properties and quality traits were reported in the organic fertilization treatments (STN, SOM, AMF, TSS, TA, lycopene content, *L**, *a**, *b**) and the differences among TP + FM and TP + CM were insignificant in their majority. On the contrary, fruit yield and its components were significantly improved in Urea + NI + UI.

## 1. Introduction

Tomato (*Solanum lycopersicum* L.) is one of the most important horticultural crops globally. Following potato, tomato is the most consumed horticultural product globally [1]. It has a high nutritional value, and namely it is rich in carotenoids, vitamin C, and antioxidants [2]. Its fruits can be consumed raw as a vegetable, or they can be processed in order to make sauces, pastes, or canned products [3]. Raw fruits represent only 20% of the total tomato production, and a significant portion of the total global tomato production corresponds to processing (or industrial) tomato [4,5,6]. Processing tomato is mainly cultivated in Italy, USA, China, Spain, Turkey, and Portugal [7]. Moreover, Greece is one of the main ten producers in Europe [8]. In 2023, the total acreage of processing tomato fields in Greece exceeded 5000 ha [9]. Being a summer crop, tomato is characterized by notable water needs, while its fertilization requirements are also relatively high [10,11]. As a result, tomato cultivation has an elevated environmental footprint [12]. The recent efforts to reduce the climate impact of agriculture have rekindled the research regarding the optimization of tomato’s agronomic management, especially its carbon footprint [12,13,14], hence assessing the optimal fertilization regime.

Nitrogen (N) is the most important nutrient for tomato’s growth and cultivation [15,16]. N fertilizers have been used for decades in order to boost crops’ development and increase yield [17]. The most common N fertilizer is urea due to its high N content and low cost [18]. Urea fertilizers are often misused and overused, leading to additional environmental degradation and significantly increasing N losses due to leaching and volatilization [17,19]. This prompted the development of more efficient N fertilizers that contain inhibitors [18]. The inhibitors in such fertilizers (slow-release fertilizers) control and slow down the release of the nutrients in soil. The most common inhibitors are the nitrification and the urease inhibitor [19].

Urease inhibitors restrict urea hydrolysis and reduce NH_3_ evaporation [20]. The most commonly used urease inhibitor is *N*-(n-butyl) thiophosphoric-triamide (NBPT) [21]. The use of NBPT can decrease N losses significantly [22]. Nitrification inhibitors suppress the conversion of ammonium (NH_4_^+^) to nitrate (NO_3_^−^) [23]. In this way, nitrification inhibitors can significantly reduce nitrate leaching [20]. Dicyandiamide (DCD) is the most common nitrification inhibitor [24]. The use of fertilizers with urease and nitrification inhibitors can cause an increase in several crops’ yield, including cotton, wheat, and maize [25,26,27,28], and reduce the climate impact of these crops by mitigating nitrous oxide and ammonia emissions [26,29]. However, more research is needed to confirm the combination of conventional fertilization at planting associated with these inhibitors for a variety of economically important crops, such as tomato for industrial processing.

Another sustainable fertilization alternative is the use of organic fertilizers and soil amendments. These products are beneficial in a number of ways, including providing a full spectrum of essential plant nutrients, adjusting pH levels to the ideal range for plant nutrient availability, increasing microbial activity and biodiversity, and enhancing the root system [30,31]. Compost derived from the plant’s waste includes both important nutrients for the plant and a source of substrate for microorganisms [32]. It is possible to find a preferable sustainable solution to all the difficulties stated above by using processing tomato waste compost as organic fertilizer.

In particular, significant quantities of tomato byproducts are created, which are known as tomato pomace [33]. The wet pomace is composed of approximately 33% seeds, 27% peel, and 40% pulp, whereas the dried pomace is composed of 44% seed and 56% pulp plus skin remaining after the fruit has been disrupted and pressed [34,35] and represents approximately 10–40% of all processed tomatoes [35,36]. The high levels of carbohydrates in pomace (approximately ranges between 25 and 50%) are believed to promote the growth of beneficial microorganisms in soils [37]. In addition, as soil quality and soil organic matter are closely related characteristics, the high level of organic matter found in processing tomato pomace will stimulate the soil quality and enhance plant growth [38]. Therefore, processed tomato pomace after the composting process can be used as organic fertilizer, reducing the environmental impact of the crop [1]. According to several studies, the use of organic fertilizers, and especially compost application (in the form of seaweed compost or tomato waste compost), can increase tomato’s yield and improve soil biological properties [1,38,39]. Within the framework of circular economy and bioeconomy models, actions such as the recycling of agricultural and livestock residues are increasingly promoted [40,41].

In view of the above, the aim of this study was to evaluate the effects of different urea fertilizers with and without urease and nitrification inhibitors and tomato pomace-based composts as organic fertilizers on the growth, yield, and quality of processing tomato (*Lycopersicon esculentum* L.) in Greece characterized by a typical semi-arid Mediterranean environment.

## 2. Results

According to the results, the use of fertilization had a positive impact on soil. The use of organic fertilizers resulted in a significant increase in soil total nitrogen (STN) and soil organic matter (SOM), 20.6 and 15.9%, and 18.8 and 18% for 2018 and 2019, respectively (Table 1). Inorganic fertilizers increased STN and SOM too. Inhibited fertilizer recorded higher results than the Urea but not as high as the organic fertilizers. Except for the soil properties, fertilization affected roots too. The biggest root length density (RLD) was recorded for the inhibited fertilizers and the lowest for the control. The use of organic fertilizers resulted in a statistically significant increase in arbuscular mycorrhiza fungi (AMF) colonization. There were non-significant differences between inorganic fertilizers and control and between the two organic fertilizers. In 2019, the use of inhibited fertilizer had a positive impact on AMF colonization, but the results were not as high as in organic fertilization. AMF colonization for processing tomato pomace with farmyard manure (TP + FM) and processing tomato pomace with compost from plant residues (TP + CM) was 46.6 and 55.0% higher than control for 2018. In 2019, TP + FM and TP + CM showed 44.9 and 36.7% higher results than control.

The yield components were also affected by fertilization. Fertilizers with inhibitors showed higher results than the others. Fruit number per pant was statistically higher in the second year (Table 2). In 2019, fruit number per plant was 4.5% higher than that in 2018. The use of fertilizers increased crop yield. The highest values for both of the experimental years were recorded in the fertilizers with nitrification and urease inhibitors (Urea + NI + UI). The use of this type of fertilizer increased the yield by 70.3 and 65.8% for 2018 and 2019, respectively, compared to the control, which had a lower yield. The average fruit weight was not affected by any of the factors.

Quality traits of processing tomato were also affected by fertilization. Total soluble solids to titratable acidity ratio (TSS/TA) was statistically significantly higher in Urea for both 2018 and 2019. Inorganic fertilizers decreased fruit firmness by about 2.5 and 6% for 2018 and 2019 (Table 3). Except for fertilization, fruit firmness was different between the two years. In 2019, fruit firmness decreased by 1.55% compared to 2018. Lycopene content was statistically significantly higher in organic fertilization for both of the experimental years, although, lycopene yield was significantly higher in the inhibited fertilizers (Table 4).

Concerning color parameters, the color index (CI) and the ratio *a*/b** were statisticaly significantly higher in all treatments (excluding the untreated control), though the differences amongst these treatments were insignificant (Table 5). On the contrary, *L** was notably reduced in Urea and Urea + NI + UI both years. The highest values were reported in TP + CM both years (40.4 and 41 for years A and B, respectively). In contrast to *L**, *a** and *b** values were significantly higher in TP + FM throughout the duration of the experiment. Even though the differences in the *a** and *b** values amongst TP + FM, TP + CM, and Urea + NI + UI were insignificant, the control values were notably lower.

The principal component analysis (PCA) explained 74.2% of the total variability (Figure 1). Most of the variation was explained by the PC1 (50.9%), which discriminated TP + CM, TP + FM, Urea, and Urea + NI + UI (on the positive side) from the control (on the negative side). PC2 revealed 23.3% of the total variance and discriminated TP + CM and TP + FM (on the positive side) from Urea and Urea + NI + UI (on the negative side). TP + CM and TP + FM were located in the upper-right quarter and were more correlated with AMF colonization, STN, SOM, TA, TSS, lycopene content, fruit diameter, *a**, and *a*/b**. Urea and Urea + NI + UI were identified in the lower-right quarter, together with CI, lycopene yield, fruit number, dry weight, fruit weight, fruit yield, RDL, and TSS/TA. Finally, the control was identified in the lower-left quarter and did not correlate with any trait.

## 3. Discussion

Based on our observations, TP + FM and TP + CM increased STN and SOM by 17–21% and 15–20%, respectively. In a study by Kakabouki et al. [38], the combined application of TP and FM and TP and CM in processing tomato increased STN and SOM by 23–30% and 11–23%, respectively. The differences in the findings of Kakabouki et al. [38] and the present study can be attributed to the different soil properties amongst the sites [42], the fertilization rates, and the composition of the organic fertilizers/soil amendments. The beneficial effects of tomato pomace, farmyard manure, and plant-residue compost on STN and SOM have been investigated in several studies that agree with our findings [38,43]. Overall, the application of such organic fertilizers and soil amendments have been proven to increase SOM [44,45]. The literature also suggests that they increase the soil microbial biomass [46] and boost their activity [47,48], thus affecting C and N decomposition [49] and their availability in the soil. Interestingly, this also coincides with the strong positive correlation between STN, SOM, and AMF colonization reported by the present study (Figure 2).

As tomatoes naturally form symbiotic relationships with AMF [50,51], substantial colonization was reported in all treatments, including the control (Table 1). However, AMF colonization was significantly higher only in TP + FM and TP + CM. The use of organic fertilizers and soil amendments has been proven to benefit AMF colonization, in contrast to the application of synthetic fertilizers [52]. For instance, in a study by Pasković et al. [53], the application of 400 kg/ha NPK 15:15:15 decreased AMF colonization by approximately 28% in tomato.

According to Liu et al. [54], plants that absorb sufficient nutrients directly (following the application of synthetic fertilizers) might reduce their reliance on AMF symbioses, hence, in such instances, the AMF colonization can decrease. On the contrary, the gradual nutrient release in organic fertilization [39,55], and particularly N [56] and P [57], could possibly benefit the AMF. This hypothesis complies with studies that suggest that high N and P concentrations could reduce AMF colonization via reducing the external hyphae formation or depressing the infectivity of the mycorrhizal propagules [51,58,59]. AMF colonization has also been reported to increase root length, density, and diameter in some vegetables, though these findings were observed under water stress conditions [60]. The enhanced RLD in the TP + FM and TP + CM treatments was probably the result of an improved nutrient uptake rather than the effect of AMF on the roots of the plants. Similar findings have also been published by Felföldi et al. [51] and Sainju et al. [55], who identified a positive correlation between fertilization and root development in tomato.

The fertilization-induced improved nutrient uptake could also explain the superior performance that was noted in all treatments (excluding the control). In addition to fruit diameter, the biomass, the yield, and its components were all significantly increased in the Urea, Urea + NI + UI, TP + FM, and TP + CM treatments. Notably, these traits were positively correlated to STN, SOM, and RDL, implying that root development and nutrient availability could be responsible for these results. Similar findings have been reported by Kakabouki et al. [38] in a study regarding the effects of tomato pomace as a soil amendment in the performance of tomato. In a study by Ddamulira et al. [61], the authors reported that soil amendments increased nutrient availability and doubled the number of fruits per plant and the yield of tomato. Felföldi et al. [51] highlighted the importance of N in tomato canopy development, as they concluded there is a positive linear correlation among N uptake and plant biomass. Overall, numerous physiological and metabolic processes are closely linked to nitrogen availability in tomato [16]. In a global meta-analysis by Cheng et al. [16], the authors concluded that N uptake can significantly affect tomato yield, vitamin C content, sugar/acid ratio, soluble sugar, and total soluble solids.

In the same study [16], the authors suggested that N fertilization rates ranging between 236 and 354 kg N ha^−1^ can increase tomato yields and TSS up to 60% and 12%, respectively, yet they could reduce lycopene content by almost 11%. This partially agrees with our findings. In the present study, similar fertilization rates (200 kg N ha^−1^) increased the yield by 65–70% and the TSS by 7–9%; however, they did not report a negative impact on the lycopene content and yield. In fact, all fertilization treatments significantly increased lycopene, with TP + FM and TP + CM resulting in the most notable increment of its content and Urea + NI + UI noting the highest lycopene yields. According to the available literature, even though small increases in the N application rates might enhance the biosynthesis of lycopene, its concentration in the tomato fruits is considered to be negatively correlated with N fertilization (decreasing N supply increases its content) [61,62,63,64,65,66]. On the contrary, K has been proposed to have a more significant (yet also genotype-dependent) effect on lycopene in tomato [67]. As both FM and CM are known for enriching soil with K [68,69], perhaps this could explain the increased lycopene content recorded in TP + FM and TP + CM. Concurrently, the higher lycopene yields in Urea + NI + UI could be attributed to the slightly superior number of fruits per plant, fruit weight, and the overall higher fruit yield that compensated for the lower content (compared to TP + FM and TP + CM).

The uptake of N does not solely affect the TSS and the lycopene content, but it can also alter the fruit firmness and the TA (thereby also the ratio TA/TSS). In most studies, both the firmness and the TA are believed to decrease with the increasing N supply [63,70], yet these findings are often contradicted [4,71]. This probably indicates that these traits depend on multiple factors (and their interactions) besides N availability and uptake. For instance, the firmness of tomato is regulated by the accumulation of Ca in the fruit, as it modulates cell wall integrity in the epidermal layer cells [72,73]. Increasing N uptake benefits the firmness up to a point. Following that crucial point, excessive N concentration in the fruits hinder the translocation of Ca [70]. This could justify the results of the present study, where TP + FM and TP + CM that release N at slower rates reported firmer fruits in comparison to Urea and Urea + NI + UI. Regarding TA, the literature points that deficit irrigation and soil moisture can significantly increase it [74,75]. As TP has been suggested to improve soil porosity [38], and improved soil porosity has been proven to increase the water-holding capacity of soil and the availability of water [76], TA was anticipated to decrease in TP + FM and TP + CM. Surprisingly, these treatments reported the highest TA values. Similar findings were recorded by Sibomana et al. [77], who observed a positive correlation among soil moisture and TA. This could possibly be ascribed to the influence of genotype on TA [78,79].

Lastly, the improved fruit surface color in TP + FM and TP + CM complies with the available literature. Besides *a*/b** and CI that did not differ significantly amongst the organic and synthetic fertilization treatments, the rest of the surface color measurement values were significantly higher in TP + FM and TP + CM. Similar findings were also reported by Kakabouki et al. [38] and by Bilalis et al. [4]. Interestingly, our work contradicts the findings that reported a negative correlation between lycopene content and *L** (Figure 2) but is in accordance with the work of Kakabouki et al. [38]. Overall, the improved *a**, *b**, and *L** were anticipated due to higher lycopene content in these treatments [4,80].

## 4. Materials and Methods

### 4.1. Site and Experimental Design

The field experiment was conducted over a period of two spring–summer cropping seasons (2018 and 2019) in the Larissa region, Central Greece (Latitude: 39°31′ N, Longitude: 22°40 ′ E, Elevation: 53 m above sea level). The soil was a Cambisol [81] and the soil structure (at 0–25 cm soil depth) was clay loam (39.3% clay, 24.6% silt, and 36.1% sand) with pH (1:1 H_2_O) 7.43, soil total nitrogen (N) 0.107%, available phosphorus (P) 42.13 mg kg^−1^ soil, available potassium (K) 257.23 mg kg^−1^ soil, 39.9% CaCO_3_, and 2.12% organic matter.

The weather data (mean monthly air temperature and precipitation) for the experimental site over the experimental periods were collected from an automatic weather station of the Institute for Environmental Research, National Observatory of Athens [82] located 11 km away; the data are shown in Figure 3. The study region has a Mediterranean climate, with hot, dry summers and cold, damp winters. The total precipitation in 2018 and 2019 (April to August) was 204.2 and 139.6 mm, respectively. The average temperature over the experiment periods was 23.1 °C in 2018 and 22.6 °C in 2019.

The experiment was set up on an area of 594 m^2^ according to a randomized complete block design (RCBD), with five fertilization treatments: control (untreated), urea (200 kg N ha^−1^), urea with nitrification and urease inhibitors (Urea + NI + UI) at a rate of 200 kg N ha^−1^, processing tomato pomace with manure (TP + FM: 50% tomato pomace + 50% farmyard manure) at a rate of 3000 kg ha^−1^, and processing tomato pomace with compost from plant residues (TP + CM: 50% tomato pomace + 50% compost from plant residue (50% wheat straw + 50% maize straw)) at a rate of 3000 kg ha^−1^ and four replications for each treatment. The urea fertilizer was 46-0-0. For the urea fertilizer with double inhibitors (46-0-0), the nitrification inhibitor (NI) was *N*-((3(5)-methyl-1H-pyrazol-1-yl) methyl) acetamide (MPA; 0.07%), whereas the urease inhibitor (UI) was *N*-(2-Nitrophenyl) phosphoric triamide (2-NPT; 0.035%). Table 6 shows the parameters of tomato pomace composts employed in the current investigation. Fertilizers were applied as a basal dressing one day before transplanting by hand spreading and then harrowing into the soil (Table 7). The amount of each type of fertilizer employed in the present experiment is the general recommended dose of the corresponding type of fertilizer (especially for tomato pomace composts) for processing tomato crop cultivation in clay-loam soils [1,4,16,38].

The plot size was 18 m^2^ (6 m × 3 m). There was a space of 1 m between plots. One day before transplanting, the fertilizers were applied by hand to the soil surface and then harrowed in. Transplanting of tomato seedlings into the field was performed on 25 April 2018 and 26 April 2019. The tested plant was tomato (*Lycopersicon esculentum* Mill.) cultivar ‘Heinz 3402 F_1_’ (Sandros SA, Thessaloniki, Greece), a ground-culture hybrid tomato variety suitable for machine harvest, with oval-square fruits of good size and consistency. It gives excellent yields in both dry and wet field conditions and is tolerant to *Verticillium* sp., *Fusarium* sp., *Meloidogyne* sp., and *Pseudomonas syringae.* Furthermore, the hybrid is well preserved due to the characteristic of prolonged field maintenance. In the present study, tomato seedlings were transplanted by hand in single rows 60 cm apart. Transplants were set at 60 cm between each other (Figure 4). The total quantity of water applied through drip irrigation during the experiment was 791 and 944 mm in 2018 and 2019, respectively. During the cropping periods, there was no incidence of pests or disease in the processing tomato crop due to the use of plant protection products. Specifically, in the present study, all pesticides were applied with a plastic backpack sprayer to the experimental plots. Pesticide treatments used in both growing seasons included two products containing systemic fungicides, Ortiva Top 20/12.5 SC (azoxystrobin 20% and difenoconazole 12.5%) and Switch 25/37.5 WG (cyprodinil 37.5% and fludioxonil 25%), one contact fungicide of Coprantol Duo 28% WG (metal cooper 28% from 14% tetraramic oxychloride and 14% hydroxide). Two insecticides, Ampligo 150 ZC (chlorantraniliprole 10% and lambda-cyhalothrin 5%) and Affirm 0.95 SG (emamectin benzoate 0.95%) (Syngenta Hellas Single Member S.A.C.I., Anthoussa Attikis, Athens, Greece), also were applied to all plots. Products containing systemic fungicides were applied once and alternately at 15 (Ortiva Top) and 40 DAT (Switch). Contact fungicide and the insecticides were each applied twice at 25 and 55 DAT. The spray regimen followed label recommendations. Moreover, weeds were controlled by hand-hoeing when needed and before canopy closure.

### 4.2. Sampling, Measurements and Methods

To analyze soil measures, two samples of topsoil (0–30 cm) were obtained from each plot 90 days after transplanting (DAT). After removing any debris, roots, and stones with a 2 mm square-hole sieve, the soil samples were air-dried at room temperature (25 °C) and kept for analysis of soil organic matter (SOM) and soil total nitrogen (STN). These measurements were conducted using the wet oxidation technique of Walkley and Black [83] and the Kjeldahl method [84], with the assistance of the Büchi B-316 device (Buchi Labortechnik AG, Flawil, Switzerland) for the combustion and extraction of the soil sample.

Root samples were also collected from the 0–35 cm layer by using a cylindrical material corer (25 cm long and 10 cm wide) at the midpoint between two plants within a row, while avoiding border plants. Specifically, three samples per plot were analyzed at 90 DAT. These samples were soaked overnight in 0.5% sodium hexametaphosphate (30 mL) solution to assist soil dispersion and root separation. The samples were then stirred for five minutes and then washed through a 5 mm mesh sieve. In this manner, roots were held on the sieves and were rinsed with tap water before staining in an acid fuchsin-lactic acid solution using the method of Kormanik and McGraw [85]. An analysis of the proportion of root length colonized by AM fungi was made microscopically by means of gridline-intersection at magnifications of ×30–40 [86]. The root length density (RLD) was determined by putting the gathered samples on a high-resolution scanner (HP Scanjet 200 Flatbed Photo Scanner; Hewlett-Packard Inc., Palo Alto, CA, USA) using Delta–T software (Delta–T Scan ver. 2.04; Delta–T Devices Ltd., Burwell, Cambridge, UK).

The dry weight per plant was determined using four randomly selected plants from each plot at 90 DAT. The plants were dried at 64 °C in oven for 72 h, and then their weight was measured. The fruit number per plant was counted on four randomly selected plants from each plot at 108 DAT. Additionally, the average fruit weight, fruit yield, and fruit diameter were determined by manually harvesting the plants in the selected rows of the plots used for counting the fruit number per plant. The fruit diameter was determined with a Starrett EC799A-6/150 electronic digital caliper (L.S. Starrett Co., Athol, MA, USA) to an accuracy of 0.02 mm.

At 100 DAT, samples of fruits were gathered for the assessment of surface color, firmness, total soluble solids (TSS), and titratable acidity (TA) content. Three fruits from ten randomly chosen plants per plot were selected for the measurements, with thirty fruits per plot in total. The measurements of surface color were conducted using a Minolta CR-300 tristimulus colorimeter, calibrated against a white standard calibration plate (Y = 93.9; x = 0.3134; y = 0.3208). The color was recorded based on the CIE-*L***a***b** color space system (*L**: brightness ranging from 0 for black to 100 for white; *a**: chroma parameter ranging from −60 for green to +60 for red; *b**: chroma parameter ranging from −60 for blue to +60 for yellow). Two measurements were taken in the equatorial area of each fruit’s pericarp, and the average value was estimated. The *a**/*b** ratio is often used to define the brightness of red tomato fruits and their products [87]. Furthermore, the color index (CI) was computed using the following equation proposed by Jimenez-Cuesta et al. [88], which has a strong association with the visual color of the fruits.
CΙ = 1000 *a***L**^−1^*b**^−1^(1)

To determine fruit firmness, a Chatillon DFIS 10 penetrometer (John Chatillon and Sons, Inc., Greensboro, NC 27425, USA) was used to measure the force required to insert a 6.3 mm diameter conical needle into the fruit to a depth of 0.6 cm using a Chatillon TCM 201-M motorized force test stand at a constant speed of 200 mm per minute in the fruit’s equatorial region. With a sensitivity of 0.2 °Brix, a Schmidt & Haensch HR32B hand-held refractometer (Schmid & Haensch GmbH & Co., 13403 Berlin, Germany) was used to determine the total soluble solids (TSS) content at 20 °C. The titratable acidity (TA) was quantified by titrating 10 mL of the homogenate diluted tomato pulp against N/50 NaOH solution using 1% phenolphthalein (1 g phenolphthalein in 100 mL of 95% ethyl alcohol) as an indicator on two ripe fruits from two randomly chosen plants per plot. The results were represented as a percentage of citric acid [89]. The TSS/TA ripening ratio was also calculated according to the method described by Sadler and Murphy [90].

For lycopene assessment, fruit samples (five fruits per plot) were selected and weighed at harvest (108 DAT) and then homogenized in a household mixer and dehydrated for 48 h using a Lyovac GT 2 freeze-dryer (Leybold-Heraeus GmbH, Köln, Germany) under vacuum. The freeze-dried samples were crushed in a laboratory mill to produce tomato flour. Lycopene extraction was carried out using 0.5 g of flour. The extraction was performed using ultrasonic-assisted extraction (UAE) with response surface methodology (RSM) and modified as described by Bilalis et al. [4]. Lycopene content, which was expressed as mg kg^−1^ fresh weight, was calculated at 503 nm using the molar extinction coefficient of 17.2 × 10^4^ M^−1^ cm^−1^ [91,92]. The lycopene yield was calculated by multiplying the lycopene content by the fruit yield.

### 4.3. Statistical Analysis

The experimental data were analyzed using the JMP 12 statistical software (SAS Institute Inc., Cary, NC, USA) according to the randomized complete block design (RCBD). One-way ANOVA was used to examine the significance of the data, and differences among means were separated using Tukey’s honestly significant difference (Tukey’s HSD) test. In addition, in the context of this study, principal component analysis (PCA) as well as correlation analyses were also used to describe the relationships between soil, growth parameters, yield components, and quality characteristics using Pearson’s correlation. All comparisons were made at the 5% level of significance.

## 5. Conclusions

The results of the current research work and their evaluation demonstrated that soil parameters were affected by different fertilization regimes, with the application of different tomato pomace composts (TP + FM and TP + CM) increasing the content of soil total nitrogen (STN) and organic matter (SOM) for the two consecutive years of the study. As for the root length density and AMF colonization of the root system of processing tomato plants, the highest values were observed in urea with urease and nitrification inhibitors (Urea + NI + UI) and tomato pomace composts (TP + FM and TP + CM), respectively. In addition, results demonstrated that plants of Urea + NI + UI plots exhibited the highest dry weight per plant, fruit number per plant, average fruit weight, fruit yield, as well as fruit diameter. In terms of fruit firmness, it was negatively affected by fertilization level, especially in an inorganic form. Regarding the color parameters, the *L** and *b** values were influenced by the fertilizer application, with the highest values found in tomato pomace composts treatments as well as untreated (control). The *a** value was significantly higher in the processing tomato pomace compost with farmyard manure (TP + FM) treatment and control, while the color indices (*a**/*b** and CI) were found to presented significantly higher values in control plots and not to differ among the plots received different urea fertilizers (with and without inhibitors) or tomato pomace composts (TP + FM and TP + CM). Significantly higher total soluble solids (TSS) were found in tomatoes fertilized with tomato pomace composts (TP + FM and TP + CM), which is significantly important to the processing tomato industry. Finally, the highest lycopene content was produced under tomato pomace composts (TP + FM and TP + CM) and there was a non-significant difference between the tomatoes receiving the processing tomato pomace compost with farmyard manure (TP + FM) and urea with urease and nitrification inhibitors (Urea + NI + UI) in terms of lycopene yield, making processing tomatoes fertilized with TP + FM suitable for lycopene production. In general, the use of organic fertilizers based on tomato pomace compost (TP + FM and TP + CM) improved significantly the quality traits of processing tomato; however, the application of urea with nitrification and urease inhibitors (Urea + NI + UI) resulted in better yield and yield components. Further research should be conducted, especially on the potential simultaneous application of organic and inorganic fertilizers; nonetheless tomato pomace seems suitable for improving the performance of processing tomato under Mediterranean conditions. In addition, there is a definite need to extend this study as a long-term experiment to evaluate the impact of seasonality.

## Figures and Tables

**Figure 1 plants-13-01852-f001:**
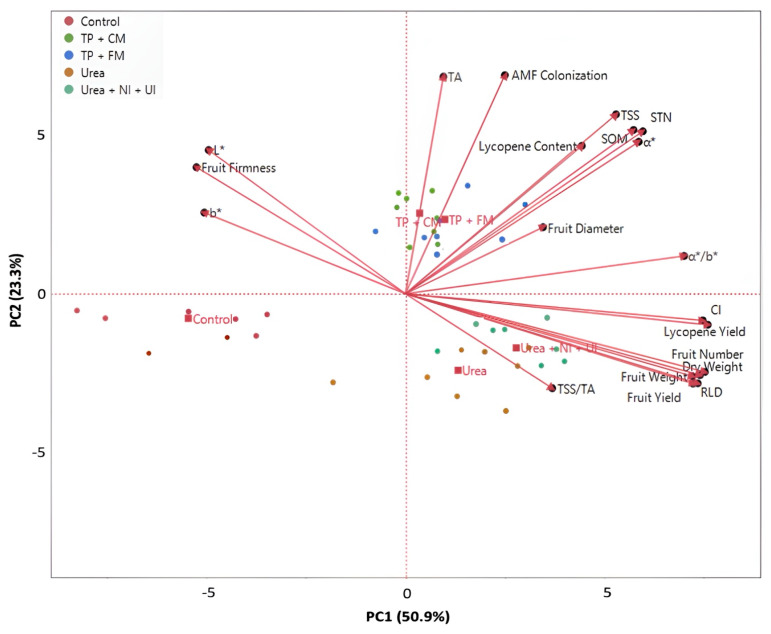
Biplot of principal component analysis results. The assessed treatments (circles) are red = control (untreated); brown = urea (200 kg N ha^−1^); teal = urea with nitrification and urease inhibitors (Urea + NI + UI) at a rate of 200 kg N ha^−1^; blue = processing tomato pomace with manure (TP + FM: 50% tomato pomace + 50% farmyard manure) at a rate of 3000 kg ha^−1^; green = processing tomato pomace with compost from plant residues (TP + CM: 50% tomato pomace + 50% compost from plant residue (50% wheat straw + 50% maize straw)) at a rate of 3000 kg ha^−1^. The studied parameters (black dots) are as follows: STN = soil total nitrogen; SOM = soil organic matter; RDL = root length density; AMF colonization = % colonization of arbuscular mycorrhizal fungi in the roots; dry weight = plant biomass; fruit number = number of tomato fruits; fruit weight = average fresh weight of the tomato fruits; fruit yield = total fruit yield; fruit firmness = average firmness of the tomato fruits; TSS = total soluble solids; TA = titratable acidity; TSS/TA = the ratio of total soluble solids to titratable acidity; lycopene content = average content of the tomato fruits in lycopene; lycopene yield = total lycopene yield; *L** = brightness of the fruits; *a** = color parameter *a*; *b** = color parameter *b*; *a**/*b** = ratio of the two chroma parameters; CI = color index.

**Figure 2 plants-13-01852-f002:**
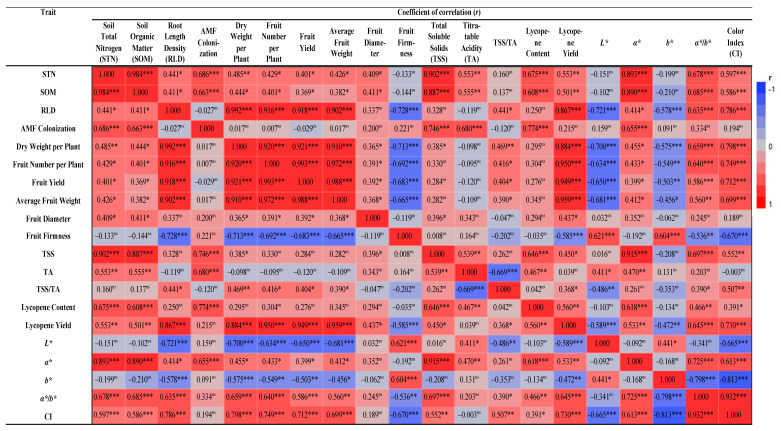
Correlation matrix between soil total nitrogen (STN), soil organic matter (SOM), root length density (RDL), AMF colonization, dry weight per plant, fruit number per plant, fruit yield, average fruit weight, fruit diameter, fruit firmness, total soluble solids (TSS), titratable acidity (TA), TSS/TA, lycopene content, lycopene yield, *L**, *a**, *b**, *a*/b**, and color index (CI). Significance levels: * *p* < 0.05; ** *p* < 0.01; *** *p* < 0.001; ns, not significant (*p* > 0.05).

**Figure 3 plants-13-01852-f003:**
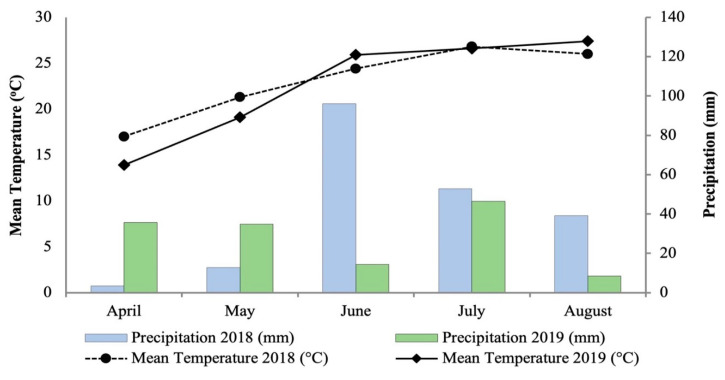
Meteorological data for the experimental location during the growing periods (April–August 2018 and 2019).

**Figure 4 plants-13-01852-f004:**
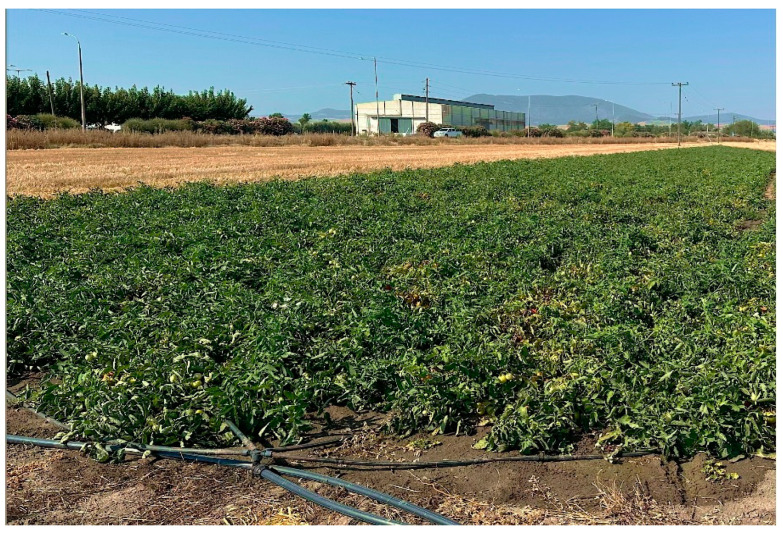
Overview of processing tomato experimental field during the second (2019) experimental year.

**Table 1 plants-13-01852-t001:** Soil total nitrogen content, soil organic matter, root length density, and arbuscular mycorrhizal fungi (AMF) colonization as affected by the different urea fertilizers and tomato pomace composts.

	Soil Total Nitrogen (STN) (%)	Soil Organic Matter (SOM) (%)	Root Length Density (RLD) (cm cm^−3^)	AMF Colonization (%)
Year A				
Control	0.107 ± 0.001 c	2.174 ± 0.023 c	0.572 ± 0.155 c	27.97 ± 1.07 b
Urea	0.116 ± 0.004 bc	2.264 ± 0.085 bc	1.479 ± 0.178 ab	31.61 ± 0.98 b
Urea + NI + UI	0.124 ± 0.006 ab	2.434 ± 0.117 ab	1.640 ± 0.167 a	30.33 ± 1.53 b
TP + FM	0.130 ± 0.003 a	2.537 ± 0.045 a	1.204 ± 0.073 ab	41.00 ± 2.45 a
TP + CM	0.128 ± 0.002 a	2.504 ± 0.058 a	1.096 ± 0.176 b	43.36 ± 1.26 a
Fertilization (F)	**	*	**	**
Blocks	ns	*	ns	ns
Year B				
Control	0.109 ± 0.002 d	2.162 ± 0.047 c	0.449 ± 0.082 d	28.89 ± 0.59 c
Urea	0.121 ± 0.004 c	2.363 ± 0.049 b	1.724 ± 0.035 ab	30.39 ± 0.74 bc
Urea + NI + UI	0.123 ± 0.001 bc	2.447 ± 0.023 ab	1.996 ± 0.090 a	31.72 ± 0.35 b
TP + FM	0.131 ± 0.003 a	2.576 ± 0.034 a	1.269 ± 0.037 bc	41.85 ± 0.71 a
TP + CM	0.128 ± 0.003 ab	2.527 ± 0.049 a	1.039 ± 0.019 c	39.55 ± 1.42 a
Fertilization (F)	***	***	***	***
Blocks	ns	ns	ns	ns
Overall effects				
Fertilization (F)	***	***	***	***
Year (Y)	ns	ns	ns	ns
F × Y	ns	ns	ns	ns
Blocks	ns	*	ns	ns

*F*-test ratios originated from ANOVA. ns: non-significant; *, **, and ***: significant at the 5%, 1%, and 0.1% levels, respectively. Values are means ± standard error. The different lowercase letters for the same year indicate significant differences among different fertilization treatments according to the Tukey’s HSD test (*p* ≤ 0.05). Control: untreated; Urea (200 kg N ha^−1^); Urea + NI + UI: urea with nitrification and urease inhibitors at a rate of 200 kg N ha^−1^; TP + FM: 50% processing tomato pomace + 50% farmyard manure at a rate of 3000 kg ha^−1^; TP + CM: 50% processing tomato pomace + 50% compost from plant residue (50% wheat straw + 50% maize straw) at a rate of 3000 kg ha^−1^.

**Table 2 plants-13-01852-t002:** Dry weight per plant, number of fruits per plant, total fruit yield, average fruit weight, and average fruit diameter as affected by the different urea fertilizers and tomato pomace composts.

	Dry Weight per Plant (g)	Fruit Number per Plant	Fruit Yield (t ha^−1^)	Average Fruit Weight (g)	Fruit Diameter (mm)
Year A					
Control	95.9 ± 5.0 c	44.0 ± 2.6 c	77.7 ± 8.1 c	47.0 ± 1.7 c	38.2 ± 0.7 a
Urea	151.2 ± 8.7 ab	53.4 ± 0.7 ab	115.9 ± 2.9 ab	58.2 ± 1.8 ab	38.5 ± 1.0 a
Urea + NI + UI	156.9 ± 8.2 a	56.9 ± 1.5 a	132.3 ± 5.9 a	62.3 ± 1.5 a	39.8 ± 1.4 a
TP + FM	137.2 ± 4.2 ab	49.6 ± 0.9 b	101.0 ± 3.5 b	54.6 ± 1.0 b	40.2 ± 0.8 a
TP + CM	131.6 ± 8.8 b	49.7 ± 0.8 b	102.1 ± 3.3 b	55.1 ± 1.1 b	39.9 ± 0.9 a
Fertilization (F)	**	**	***	***	ns
Blocks	ns	ns	ns	ns	ns
Year B					
Control	93.2 ± 5.4 d	45.3 ± 1.4 d	84.5 ± 10.2 d	49.7 ± 3.3 d	39.1 ± 0.6 a
Urea	163.3 ± 1.7 a	55.8 ± 0.9 ab	122.9 ± 2.7 ab	59.1 ± 0.5 ab	39.2 ± 0.7 a
Urea + NI + UI	174.4 ± 4.6 a	59.6 ± 1.7 a	140.1 ± 6.3 a	63.0 ± 1.1 a	40.9 ± 0.9 a
TP + FM	140.2 ± 2.5 b	53.8 ± 0.8 bc	113.4 ± 3.8 bc	56.6 ± 0.9 bc	40.6 ± 0.8 a
TP + CM	128.9 ± 1.9 c	50.6 ± 1.2 c	100.4 ± 4.7 cd	53.2 ± 0.8 cd	40.2 ± 1.1 a
Fertilization (F)	***	***	***	**	ns
Blocks	ns	ns	ns	ns	ns
Overall effects					
Fertilization (F)	***	***	***	***	ns
Year (Y)	ns	*	ns	ns	ns
F × Y	ns	ns	ns	ns	ns
Blocks	ns	ns	ns	ns	ns

*F*-test ratios originated from ANOVA. ns: non-significant; *, **, and ***: significant at the 5%, 1%, and 0.1% levels, respectively. Values are means ± standard error. The different lowercase letters for the same year indicate significant differences among different fertilization treatments according to the Tukey’s HSD test (*p* ≤ 0.05). Control: untreated; Urea (200 kg N ha^−1^); Urea + NI + UI: urea with nitrification and urease inhibitors at a rate of 200 kg N ha^−1^; TP + FM: 50% processing tomato pomace + 50% farmyard manure at a rate of 3000 kg ha^−1^; TP + CM: 50% processing tomato pomace + 50% compost from plant residue (50% wheat straw + 50% maize straw) at a rate of 3000 kg ha^−1^.

**Table 3 plants-13-01852-t003:** Fruit firmness, total soluble solids (TSS) content, titratable acidity (TA) content, and TSS/TA ratio as affected by the different urea fertilizers and tomato pomace composts.

	Fruit Firmness (kg cm^−2^)	Total Soluble Solids (TSS) (°Brix)	Titratable Acidity (TA) (% Citric Acid *w*/*w*)	TSS/TA
Year A				
Control	4.59 ± 0.02 a	4.49 ± 0.08 c	0.29 ± 0.02 bc	15.35 ± 0.53 bc
Urea	4.45 ± 0.03 b	4.74 ± 0.10 bc	0.27 ± 0.01 c	17.34 ± 0.29 a
Urea + NI + UI	4.48 ± 0.01 b	4.82 ± 0.02 b	0.29 ± 0.01 b	16.45 ± 0.48 ab
TP + FM	4.57 ± 0.03 a	5.11 ± 0.12 a	0.32 ± 0.03 a	16.20 ± 0.30 abc
TP + CM	4.52 ± 0.02 ab	5.01 ± 0.07 ab	0.33 ± 0.02 a	15.09 ± 0.51 c
Fertilization (F)	**	**	***	*
Blocks	ns	ns	*	ns
Year B				
Control	4.58 ± 0.09 a	4.47 ± 0.06 d	0.29 ± 0.01 c	15.39 ± 0.49 b
Urea	4.34 ± 0.05 bc	4.81 ± 0.07 c	0.28 ± 0.02 c	17.32 ± 0.18 a
Urea + NI + UI	4.28 ± 0.02 c	4.87 ± 0.05 bc	0.30 ± 0.01 bc	16.50 ± 0.21 ab
TP + FM	4.56 ± 0.08 ab	5.23 ± 0.08 a	0.31 ± 0.02 ab	16.54 ± 0.82 ab
TP + CM	4.50 ± 0.03 ab	5.11 ± 0.03 ab	0.33 ± 0.01 a	15.40 ± 0.54 b
Fertilization (F)	*	***	**	*
Blocks	*	ns	ns	ns
Overall effects				
Fertilization (F)	***	***	***	**
Year (Y)	*	ns	ns	ns
F × Y	ns	ns	ns	ns
Blocks	*	ns	*	ns

*F*-test ratios originated from ANOVA. ns: non-significant; *, **, and ***: significant at the 5%, 1%, and 0.1% levels, respectively. Values are means ± standard error. The different lowercase letters for the same year indicate significant differences among different fertilization treatments according to the Tukey’s HSD test (*p* ≤ 0.05). Control: untreated; Urea (200 kg N ha^−1^); Urea + NI + UI: urea with nitrification and urease inhibitors at a rate of 200 kg N ha^−1^; TP + FM: 50% processing tomato pomace + 50% farmyard manure at a rate of 3000 kg ha^−1^; TP + CM: 50% processing tomato pomace + 50% compost from plant residue (50% wheat straw + 50% maize straw) at a rate of 3000 kg ha^−1^.

**Table 4 plants-13-01852-t004:** Lycopene content and lycopene yield as affected by the different urea fertilizers and tomato pomace composts.

	Lycopene Content (mg kg^−1^ Fresh Weight)	Lycopene Yield (kg ha^−1^)
Year A		
Control	74.4 ± 4.2 c	5.9 ± 1.0 c
Urea	80.3 ± 1.2 bc	9.3 ± 0.2 ab
Urea + NI + UI	81.9 ± 1.8 ab	10.8 ± 0.6 a
TP + FM	88.0 ± 0.3 a	8.9 ± 0.3 b
TP + CM	88.8 ± 1.9 a	9.1 ± 0.5 ab
Fertilization (F)	**	**
Blocks	ns	ns
Year B		
Control	74.6 ± 3.9 c	6.4 ± 0.8 c
Urea	84.2 ± 1.4 ab	10.4 ± 0.4 ab
Urea + NI + UI	78.6 ± 2.9 bc	11.0 ± 0.5 a
TP + FM	86.9 ± 0.7 a	9.8 ± 0.4 ab
TP + CM	88.2 ± 2.4 a	8.8 ± 0.3 b
Fertilization (F)	*	**
Blocks	ns	ns
Overall effects		
Fertilization (F)	**	***
Year (Y)	ns	ns
F × Y	ns	ns
Blocks	ns	ns

*F*-test ratios originated from ANOVA. ns: non-significant; *, **, and ***: significant at the 5%, 1%, and 0.1% levels, respectively. Values are means ± standard error. The different lowercase letters for the same year indicate significant differences among different fertilization treatments according to the Tukey’s HSD test (*p* ≤ 0.05). Control: untreated; Urea (200 kg N ha^−1^); Urea + NI + UI: urea with nitrification and urease inhibitors at a rate of 200 kg N ha^−1^; TP + FM: 50% processing tomato pomace + 50% farmyard manure at a rate of 3000 kg ha^−1^; TP + CM: 50% processing tomato pomace + 50% compost from plant residue (50% wheat straw + 50% maize straw) at a rate of 3000 kg ha^−1^.

**Table 5 plants-13-01852-t005:** Fruit surface color (color parameters *L**, *a*,* and *b**, *a**/*b** ratio, and the color index (CI)) as affected by the different urea fertilizers and tomato pomace composts.

	Fruit Surface Color				
	*L**	*a**	*b**	*a**/*b**	CI
Year A					
Control	40.8 ± 0.5 a	33.3 ± 0.4 c	27.8 ± 0.2 a	1.20 ± 0.01 b	29.3 ± 0.6 b
Urea	39.0 ± 0.2 c	34.4 ± 0.7 bc	26.9 ± 0.1 c	1.27 ± 0.03 a	32.7 ± 1.1 a
Urea + NI + UI	39.3 ± 0.4 bc	35.0 ± 0.1 ab	27.1 ± 0.3bc	1.29 ± 0.01 a	32.9 ± 0.5 a
TP + FM	40.3 ± 0.3 ab	36.1 ± 0.8 a	27.8 ± 0.2 a	1.30 ± 0.02 a	32.3 ± 0.6 a
TP + CM	40.4 ± 0.4 ab	35.4 ± 0.4 ab	27.5 ± 0.1 ab	1.28 ± 0.01 a	31.8 ± 0.2 a
Fertilization (F)	*	*	**	*	*
Blocks	ns	ns	ns	ns	ns
Year B					
Control	41.4 ± 0.3 a	33.5 ± 0.3 b	27.9 ± 0.9 a	1.21 ± 0.05 b	29.1 ± 1.3 b
Urea	39.1 ± 0.1 b	34.7 ± 0.5 ab	25.5 ± 0.4 c	1.36 ± 0.01 a	34.7 ± 0.4 a
Urea + NI + UI	39.5 ± 0.2 b	35.3 ± 0.7 a	25.7 ± 0.6 bc	1.38 ± 0.03 a	34.9 ± 1.2 a
TP + FM	40.7 ± 0.3 a	36.2 ± 0.6 a	27.2 ± 0.3 ab	1.33 ± 0.02 a	32.7 ± 0.7 a
TP + CM	41.0 ± 0.4 a	35.7 ± 0.2 a	25.9 ± 0.1 bc	1.38 ± 0.01 a	33.5 ± 0.2 a
Fertilization (F)	**	*	*	*	**
Blocks	ns	ns	ns	ns	ns
Overall effects					
Fertilization (F)	**	**	**	***	***
Year (Y)	ns	ns	***	**	*
F × Y	ns	ns	ns	ns	ns
Blocks	ns	ns	ns	ns	ns

*F*-test ratios originated from ANOVA. ns: non-significant; *, **, and ***: significant at the 5%, 1%, and 0.1% levels, respectively. Values are means ± standard error. The different lowercase letters for the same year indicate significant differences among different fertilization treatments according to the Tukey’s HSD test (*p* ≤ 0.05). Control: untreated; Urea (200 kg N ha^−1^); Urea + NI + UI: urea with nitrification and urease inhibitors at a rate of 200 kg N ha^−1^; TP + FM: 50% processing tomato pomace + 50% farmyard manure at a rate of 3000 kg ha^−1^; TP + CM: 50% processing tomato pomace + 50% compost from plant residue (50% wheat straw + 50% maize straw) at a rate of 3000 kg ha^−1^.

**Table 6 plants-13-01852-t006:** Chemical properties of tomato pomace composts used in the present study.

	Organic Matter (%)	EC (mS cm^−1^)	pH	N Total (g kg^−1^)	P Olsen (mg kg^−1^)	K (mg kg^−1^)	Mg (mg kg^−1^)
Tomato pomace with farmyard manure (TP + FM)	52	1.66	7.43	31.6	20	34	0.65
Tomato pomace with compost (TP + CM)	44	1.76	7.38	28.7	15	30	0.35

**Table 7 plants-13-01852-t007:** Fertilization amount, nitrogen (N) content, and N application rate for each fertilization treatment used in the present study.

Fertilization Treatment	Fertilization Amount	Nitrogen (N) Content	N Application Rate
Control	No fertilizer	-	-
Urea	435 kg ha^−1^	46%	200 kg N ha^−1^
Urea + NI + UI	435 kg ha^−1^	46%	200 kg N ha^−1^
TP + FM	3000 kg ha^−1^	3.16%	94.8 kg N ha^−1^
TP + CM	3000 kg ha^−1^	2.87%	86.1 kg N ha^−1^

Control: untreated; Urea; Urea + NI + UI: urea with nitrification and urease inhibitors; TP + FM: 50% processing tomato pomace + 50% farmyard manure; TP + CM: 50% processing tomato pomace + 50% compost from plant residue (50% wheat straw + 50% maize straw).

## Data Availability

All data generated or analyzed during this study are included in this published article. Further inquiries can be addressed to the corresponding author.

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
