# Peer review of "Comparative Study Effect of Different Urea Fertilizers and Tomato Pomace Composts on the Performance and Quality Traits of Processing Tomato (Lycopersicon esculentum Mill.)"

_plants, 2024, doi:10.3390/plants13131852_

Round 1
Reviewer 1 Report
Comments and Suggestions for Authors
The introduction should be improved with the most recent published work on processing tomato using organic fertilizers, climate change and irrigation.
I suggest to improve table 6 or add another one reporting all investigated treatments and reporting the total amount of kg of N used per hectare
Please report the plant density used for this study, 60 cm between single rows?
No incidence of pests? Really?
How was the irrigation water supplied? Which system was used?
Line 123, only three fruits? In my opinion are not representative of the plots, we use at least 100 fruits in our experiments
Conclusions should be rewritten adding the limitation of the study
In the statistical analysis, any effects of the blocks?
Figure and table should contain the full description of all abbreviations, in the captions
Author Response
General Response: We thank the reviewer for the positive comments on our study and for the interesting information useful for improving it. We also double-checked the language of the text.
The introduction should be improved with the most recent published work on processing tomato using organic fertilizers, climate change and irrigation.
Response: We revised according to the reviewer's request. The changes were marked up using the “Track Changes” function. (“Another sustainable fertilization alternative is the use of organic fertilizers and soil amendments. …….. Therefore, processing tomato pomace after composting process can be used as organic fertilizer, reducing the environmental impact of the crop [1].”).
I suggest to improve table 6 or add another one reporting all investigated treatments and reporting the total amount of kg of N used per hectare.
Response: We revised according to the reviewer's recommendations. We added another one Table (Table 7) entitled: “Fertilization amount, nitrogen (N) content, and N application rate for each fertilization treatment used in the present study”. In addition, we referred that: "the amount of each type of fertilizer employed in the present experiment is the general recommended dose of the corresponding type of fertilizer (especially for tomato pomace composts) for processing tomato crop cultivation in clay-loam soils [1,4,16,38]."
Please report the plant density used for this study, 60 cm between single rows?
Response: We revised according to the reviewer's request. The changes were marked up using the “Track Changes” function. (“In the present study, tomato seedlings were transplanted by hand in single rows 60 cm apart.”)
No incidence of pests? Really?
Response: We revised according to the reviewer's request. The changes were marked up using the “Track Changes” function. (“During the cropping periods, there was no incidence of pests or disease in the processing tomato crop due to the use of plant protection products. Specifically, in the present study, all pesticides were applied with a plastic backpack sprayer to the experimental plots. Pesticide treatments used in both growing seasons included two products containing systemic fungicides, Ortiva Top 20/12.5 SC (azoxystrobin 20% and difenoconazole 12.5%) and Switch 25/37.5 WG (cyprodinil 37.5% and fludioxonil 25%), one contact fungicide of Coprantol Duo 28% WG (metal cooper 28% from 14% tetraramic oxychloride and 14% hydroxide). Two insecticides, Ampligo 150 ZC (chlorantraniliprole 10% and lamb-da-cyhalothrin 5%) and Affirm 0.95 SG (emamectin benzoate 0.95%) (Syngenta Hellas Single Member S.A.C.I., Anthoussa Attikis, Athens, Greece), also were applied to all plots. Products containing systemic fungicides were applied once and alternately at 15 (Ortiva Top) and 40 DAT (Switch). Contact fungicide and the insecticides were each ap-plied twice at 25 and 55 DAT. The spray regimen followed label recommendations.”)
How was the irrigation water supplied? Which system was used?
Response: We revised according to the reviewer's request. The changes were marked up using the “Track Changes” function. (“Transplants were set at 60 cm between each other (Figure 4). The total quantity of water applied through drip irrigation during the experiment was 791 and 944 mm in 2018 and 2019, respectively.”)
Line 123, only three fruits? In my opinion are not representative of the plots, we use at least 100 fruits in our experiments
Response: The correct sentence is: “Three fruits from ten randomly chosen plants per plot were selected for the measurements, with thirty fruits per plot in total”. The number of fruits per plot (30) was sufficient according to the methodology we used in our experiment.
Conclusions should be rewritten adding the limitation of the study.
Response: We revised according to the reviewer's request. The changes were marked up using the “Track Changes” function. (“The results of the current research work and their evaluation demonstrated that soil parameters were affected by different fertilization regimes, with the application of different tomato pomace composts (TP + FM and TP + CM) increased the content of soil total nitrogen (STN) and organic matter (SOM) for the two consecutive years of the study. …. In addition, there is a definite need to extend this study as a long-term experiment to evaluate the impact of seasonality.”)
In the statistical analysis, any effects of the blocks?
Response: We revised according to the reviewer's recommendations. The changes were marked up using the “Track Changes” function in the Tables of Results (We added the effect of blocks)
Figure and table should contain the full description of all abbreviations, in the captions
Response: The corrections were marked up using the “Track Changes” function according to the reviewer’s comments. (“Control: untreated; Urea (200 kg N ha-1); Urea + NI + UI: urea with nitrification and urease inhibi-tors at a rate of 200 kg N ha-1; TP + FM: 50% processing tomato pomace + 50% farmyard manure at a rate of 3000 kg ha-1; TP + CM: 50% processing tomato pomace + 50% compost from plant res-idue (50% wheat straw + 50% maize straw) at a rate of 3000 kg ha-1.”)
Reviewer 2 Report
Comments and Suggestions for Authors
The manuscript presents a comprehensive study of the effects of various fertilization methods on the growth and quality of processing tomatoes. The authors have investigated traditional urea-based fertilizers and innovative organic alternatives using tomato pomace, providing valuable insights into sustainable agricultural practices. The experimental design is robust, utilizing a randomized complete block design over two years, strengthening the results' reliability.
The manuscript is generally well-written with clear objectives and a logical flow. However, the introduction could benefit from a more concise overview of the key literature to position the study more clearly within the existing research landscape
The methodology is detailed and well-structured, clearly describing the experimental setup, data collection, and statistical analysis. However, the authors should consider including a discussion on any potential limitations of their study, such as the geographical specificity of the results or the scalability of using tomato pomace in different agricultural contexts.
The manuscript is well-cited, utilizing a range of current and relevant sources. Ensuring that all recent key works are referenced would be beneficial. Additionally, some citations in the discussion could be updated to include the most recent research to strengthen the narrative.
Comments on the Quality of English LanguageThe manuscript is generally well-written.
Author Response
The manuscript presents a comprehensive study of the effects of various fertilization methods on the growth and quality of processing tomatoes. The authors have investigated traditional urea-based fertilizers and innovative organic alternatives using tomato pomace, providing valuable insights into sustainable agricultural practices. The experimental design is robust, utilizing a randomized complete block design over two years, strengthening the results' reliability.
Response: We thank the reviewer for the positive comments on our study and for the interesting information useful for improving it. We also double-checked the language of the text.
The manuscript is generally well-written with clear objectives and a logical flow. However, the introduction could benefit from a more concise overview of the key literature to position the study more clearly within the existing research landscape.
Response: We revised according to the reviewer's request. The changes were marked up using the “Track Changes” function. (“Another sustainable fertilization alternative is the use of organic fertilizers and soil amendments. …….. Therefore, processing tomato pomace after composting process can be used as organic fertilizer, reducing the environmental impact of the crop [1].”).
The methodology is detailed and well-structured, clearly describing the experimental setup, data collection, and statistical analysis. However, the authors should consider including a discussion on any potential limitations of their study, such as the geographical specificity of the results or the scalability of using tomato pomace in different agricultural contexts.
Response: As referred in the introduction and materials and methods section, the aim of this study was to evaluate the effects of different urea fertilizers with and without urease and nitrification inhibitors, and tomato pomace-based composts as organic fertilizers on the growth, yield and quality of processing tomato (Lycopersicon esculentum L.) cultivated in a clay loam soil in Greece characterized by a semi-arid Mediterranean environment. In addition, it is well known that tomatoes grown in relatively warm temperatures, and the Mediterranean environment is typical for tomato cultivation and production. Taking into account the above, we believe that the current study specifies the geographical specify of the results by providing the climate of the studied area and the meteorological data (in the material and methods section) which are typical and appropriate for tomato cultivation during the cultivation period of the study.
The manuscript is well-cited, utilizing a range of current and relevant sources. Ensuring that all recent key works are referenced would be beneficial. Additionally, some citations in the discussion could be updated to include the most recent research to strengthen the narrative.
Response: We revised according to the reviewer's request. The changes were marked up using the “Track Changes” function. Specifically, we have now added the following references:
Bamdad, H.; Papari, S.; Lazarovits, G.; Berruti, F. Soil amendments for sustainable agriculture: Microbial organic fertilizers. Soil Use Manag. 2022, 38, 94–120.
Felföldi, Z.; Vidican, R.; Stoian, V.; Roman, I.A.; Sestras, A.F.; Rusu, T.; Sestras, R.E. Arbuscular mycorrhizal fungi and fertili-zation influence yield, growth and root colonization of different tomato genotype. Plants 2022, 11, 1743.
Gao, F.; Li, H.; Mu, X.; Gao,H.; Zhang, Y.; Li, R.; Cao, K.; Ye, L. Effects of organic fertilizer application on tomato yield and qual-ity: A meta-analysis. Appl. Sci. 2023, 13, 2184.
Mukherjee, S.; Dash, P.K.; Das, D.; Das, S. Growth, yield and water productivity of tomato as influenced by deficit irrigation water management. Environ. Process. 2023, 10, 10.
Murariu, O.C.; Brezeanu,C.; Jitareanu, C.D.; Robu, T.; Irimia,L.M.; Trofin, A.E.; Popa, L.-D.;Stoleru, V.; Murariu, F.; Brezeanu, P.M. Functional quality of improved tomato genotypes grown in open field and in plastic tunnel under organic farming. Agri-culture 2021, 11, 609.
Reviewer 3 Report
Comments and Suggestions for Authors
Comments and Suggestions for Authors
Title: Comparative Study Effect of Different Urea Fertilizers and Tomato
Pomace Composts on the Performance and Quality Traits of Processing Tomato
(Lycopersicon esculentum Mill.)
Dear Authors
The manuscript presents research results that fall within the editorial scope of the Plants journal. The research problem presented in MS covers a global scope. The number of results presented is sufficient. The results are correctly summarized in subsequent tables and generally well described in the Results section. In the Discussion section, the authors compared the results of their own research to the research of other authors. The conclusions need to be improved. In order to increase the usefulness of the article, Authors must refer to the following points.
Remarks:
1. Introduction: Lines 70-71 Organic waste isn't organic fertilizer. Needs to be improved.
2. Results: I propose to remove the first part of the titles in tables 1-5 in accordance with the data presented in the tables. Indexes placed next to the data indicate that they have been subjected to statistical analysis.
3. Discussion: Figure 2 - This is a table.
4. Materials and Methods: subsection 4.1. The soil type should be provided according to the WRB (World Reference Base for Soil Resources), 4th edition, 2022. Were other mineral fertilizers used? If so, what are they? The units should be corrected: farmyard manure dose in t ha-1 or Mg ha-1; table 6 - Ntot content in g kg-1.
5. Conclusions: The conclusions should be extended and related to the parameters examined.
Best regards
Comments and Suggestions for Authors
Title: Comparative Study Effect of Different Urea Fertilizers and Tomato
Pomace Composts on the Performance and Quality Traits of Processing Tomato
(Lycopersicon esculentum Mill.)
Dear Authors
The manuscript presents research results that fall within the editorial scope of the Plants journal. The research problem presented in MS covers a global scope. The number of results presented is sufficient. The results are correctly summarized in subsequent tables and generally well described in the Results section. In the Discussion section, the authors compared the results of their own research to the research of other authors. The conclusions need to be improved. In order to increase the usefulness of the article, Authors must refer to the following points.
Remarks:
1. Introduction: Lines 70-71 Organic waste isn't organic fertilizer. Needs to be improved.
2. Results: I propose to remove the first part of the titles in tables 1-5 in accordance with the data presented in the tables. Indexes placed next to the data indicate that they have been subjected to statistical analysis.
3. Discussion: Figure 2 - This is a table.
4. Materials and Methods: subsection 4.1. The soil type should be provided according to the WRB (World Reference Base for Soil Resources), 4th edition, 2022. Were other mineral fertilizers used? If so, what are they? The units should be corrected: farmyard manure dose in t ha-1 or Mg ha-1; table 6 - Ntot content in g kg-1.
5. Conclusions: The conclusions should be extended and related to the parameters examined.
Best regards
Comments on the Quality of English LanguageMinor editing of English language required
Author Response
Dear Authors
The manuscript presents research results that fall within the editorial scope of the Plants journal. The research problem presented in MS covers a global scope. The number of results presented is sufficient. The results are correctly summarized in subsequent tables and generally well described in the Results section. In the Discussion section, the authors compared the results of their own research to the research of other authors. The conclusions need to be improved. In order to increase the usefulness of the article, Authors must refer to the following points.
Response: We thank the reviewer for the positive comments on our study and for the interesting information useful for improving it. We also double-checked the language of the text. Please see below the response to your remarks.
Remarks:
- Introduction: Lines 70-71 Organic waste isn't organic fertilizer. Needs to be improved.
Response: We revised according to the reviewer's request. The changes were marked up using the “Track Changes” function. (“Therefore, processing tomato pomace after composting process can be used as organic fertilizer, reducing the environmental impact of the crop [1].”)
- Results: I propose to remove the first part of the titles in tables 1-5 in accordance with the data presented in the tables. Indexes placed next to the data indicate that they have been subjected to statistical analysis.
Response: We revised according to the reviewer's request. The changes were marked up using the “Track Changes” function in the Titles of Result Tables (Tables 1-5).
- Discussion: Figure 2 - This is a table.
Response: We revised according to the reviewer's request. The changes were marked up using the “Track Changes” function.
- Materials and Methods: subsection 4.1. The soil type should be provided according to the WRB (World Reference Base for Soil Resources), 4th edition, 2022. Were other mineral fertilizers used? If so, what are they? The units should be corrected: farmyard manure dose in t ha-1 or Mg ha-1; table 6 - Ntot content in g kg-1
Response: We revised according to the reviewer's request. The changes were marked up using the “Track Changes” function. (“ The soil was a Cambisol [82] and the soil structure (at 0–25 cm soil depth) was clay loam (39.3% clay, 24.6% silt and 36.1% sand) with pH (1:1 H2O) 7.43,..”) Please also see Table 6.
- Conclusions: The conclusions should be extended and related to the parameters examined.
Response: We revised according to the reviewer's request. The changes were marked up using the “Track Changes” function. (“The results of the current research work and their evaluation demonstrated that soil parameters were affected by different fertilization regimes, with the application of different tomato pomace composts (TP + FM and TP + CM) increased the content of soil total nitrogen (STN) and organic matter (SOM) for the two consecutive years of the study. …. In addition, there is a definite need to extend this study as a long-term experiment to evaluate the impact of seasonality.”)
Round 2
Reviewer 1 Report
Comments and Suggestions for Authors
Authors addressed the main concerns